# Excavation of Genes Responsive to Brassinosteroids by Transcriptome Sequencing in *Adiantum flabellulatum* Gametophytes

**DOI:** 10.3390/genes13061061

**Published:** 2022-06-14

**Authors:** Zeping Cai, Zhenyu Xie, Xiaochen Wang, Shuixian Zhang, Qian Wu, Xudong Yu, Yi Guo, Shuyi Gao, Yunge Zhang, Shitao Xu, Honggang Wang, Jiajia Luo

**Affiliations:** 1Key Laboratory of Genetics and Germplasm Innovation of Tropical Special Forest Trees and Ornamental Plants, Ministry of Education, College of Forestry, Hainan University, Haikou 570228, China; 992995@hainanu.edu.cn (Z.C.); xiezhenyu2333@163.com (Z.X.); xiaochenwang2001@163.com (X.W.); znarcissu@163.com (S.Z.); guoyihainan@163.com (Y.G.); xxgaosy@163.com (S.G.); 2College of Tropical Crops, Hainan University, Haikou 570228, China; wuqian_2019@126.com (Q.W.); 20196703310025@hainanu.edu.cn (Y.Z.); deru666@hainanu.edu.cn (H.W.); 3Key Laboratory of Germplasm Resources Biology of Tropical Special Ornamental Plants of Hainan Province, College of Forestry, Hainan University, Haikou 570228, China; doeast@163.com; 4College of Horticulture, Hainan University, Haikou 570228, China; xushitao@hainanu.edu.cn; 5Tropical Crops Genetic Resources Institute, Chinese Academy of Tropical Agricultural Sciences, Haikou 571101, China

**Keywords:** ferns, photosynthesis, brassinosteroids synthesis, negative feedback regulation

## Abstract

Brassinosteroids (BRs) are a class of polyhydroxysteroid plant hormones; they play important roles in the development and stress resistance of plants. The research on BRs has mainly been carried out in angiosperms, but in ferns—research is still limited to the physiological level and is not in-depth. In this study, *Adiantum flabellulatum* gametophytes were used as materials and treated with 10^−6^ M brassinolide (BL). The differentially expressed genes (DEGs) responsive to BRs were identified by transcriptome sequencing, GO, KEGG analysis, as well as a quantitative real-time polymerase chain reaction. From this, a total of 8394 DEGs were screened. We found that the expressions of photosynthetic genes were widely inhibited by high concentrations of BL in *A. flabellulatum* gametophytes. Moreover, we detected many BR synthase genes, except *BR6ox2*, which may be why castasterone (CS) rather than BL was detected in ferns. Additionally, we identified (for the first time) that the expressions of BR synthase genes (*CYP90B1*, *CYP90C1*, *CYP90D1*, *CPD,* and *BR6ox1*) were negatively regulated by BL in fern gametophytes, which indicated that ferns, including gametophytes, also needed the regulatory mechanism for maintaining BR homeostasis. Based on transcriptome sequencing, this study can provide a large number of gene expression data for BRs regulating the development of fern gametophytes.

## 1. Introduction

Brassinosteroids (BRs) are a class of polyhydroxysteroid plant hormones. Nowadays, more than 70 analogs have been found, among which brassinolide (BL) is the most active form [1,2,3]. BRs play important roles in regulating plant growth and development, such as promoting cell elongation, regulating plant fertility, and affecting photomorphogenesis [4,5]. The BR signal is transduced by the receptor-like protein kinases BRI1/BAK1-mediated pathway, in which BRI1 and BAK1 are the receptor and co-receptor of BRs, respectively [6,7]. So far, the research on BRs has mainly been carried out in angiosperms [8], but in ferns, research has been limited to the physiological level and is not in-depth. For example, BL with concentrations of 10^−7^ M and 10^−6^ M can inhibit spore germination of *Pteridium aquilinum* and gametophyte growth of *A. flabellulatum*, respectively [9,10], but the related genes are still unknown.

Castasterone (CS) is the immediate precursor of BL [11]. At present, CS, rather than BL, has been found in 12 species of Polypodiopsida (true ferns), such as *P. aquilinum, Osmunda japonica*, and *Deparia japonica* [12]. In angiosperms, some monocots, such as *Oryza sativa* and *Zea mays,* can only synthesize CS [12], but some dicots, such as *Arabidopsis thaliana* and *Solanum lycopersicum* (tomato), can synthesize BL from CS [13,14]. The studies show that the synthesis of CS from campesterol (CR) requires many enzymes, but CS can further synthesize BL only when BR6ox2 (C-6 oxidation) is present [15,16,17]. The above results indicate that BR6ox2 is presumably not present in ferns. Additionally, plants have the regulatory mechanisms for maintaining BR homeostasis [18]. For example, the expressions of BR synthase genes (*CYP90B1* (*DWF4*), *CYP90C1* (*ROT3*), *CYP90D1*, *CPD*, *BR6ox1,* and *BR6ox2*) are negatively regulated by BRs in *A. thaliana* [19,20]. Although many BR synthase genes are present in ferns [12], it is still unknown whether their expressions are negatively regulated by BRs.

Transcriptome sequencing has been widely used in the screening of differentially expressed genes (DEGs) and pathway analysis. Huang et al. [21], for example, treated *Gerbera hybrida* petals with BL, then used transcriptome sequencing, gene ontology (GO), and the Kyoto Encyclopedia of Genes and Genomes (KEGG) analysis to identify the DEGs, which were annotated to multiple plant hormone signaling pathways. Wu et al. [22] employed the methods above and discovered that BL increased the contents of total glucosinolates and sulforaphane by upregulating the gene expressions related to glucosinolate core pathways in *Brassica oleracea*. BL is the most commonly used drug in the study of BR regulating plant development, and it is also commonly used in *O. sativa*, *Z. mays, P. aquilinum*, and other plants that only synthesize CS [9,23,24,25,26,27,28]. In view of this, 10^−6^ M BL was applied to the *A. flabellulatum* gametophytes. The genes responsive to BL were identified and the BR synthase genes negatively regulated by BL were explored through transcriptome sequencing, GO, KEGG analysis, as well as a quantitative real-time polymerase chain reaction (qRT-PCR). So, this study can provide a large number of gene expression data for BR regulating the development of fern gametophytes.

## 2. Materials and Methods

### 2.1. Experimental Materials and Drugs

*A. flabellulatum* spores were collected from the rubber forest in Ma’an Mountain, Danzhou City, Hainan, China (109.519° E, 19.501° N), then stored at 4 °C in a dry, closed container. BL was purchased from Solarbio Company (CAS:78821-43-9), prepared into a 10^−3^ M mother solution with 75% ethanol, and stored at 4 °C.

### 2.2. Experimental Methods

#### 2.2.1. Culture of *Adiantum flabellulatum* Gametophytes

*A. flabellulatum* spores were immersed in 0.1% HgCl_2_ to sterilize for 5 min and soaked in sterile water 5 times; they were then inoculated on 1/4 MS solid medium and cultured under light conditions (illumination was 500 Lx, the temperature was 24 ± 5 °C). After germination, young gametophytes with diameters of 2 mm were selected and transferred to a new 1/4 MS solid medium with different BL concentrations (0, 10^−12^, 10^−11^, 10^−10^, 10^−9^, 10^−8^, 10^−7^, and 10^−6^ M), then cultured for 60 d under the illumination of 1000 Lx (24 h/d, 24 ± 2 °C). Nine biological replicates were set for each of the treatments. After photography, the perimeters and the projected areas of gametophytes in each treatment were measured by Image J (v1.8.0, Bethesda, Rockville, MD, USA), then the means and standard deviations were counted, and statistical significance analyses were performed by a least significant difference (LSD) test (SPSS 25.0, Armonk, NY, USA).

#### 2.2.2. RNA Extraction, Library Construction, Sequencing, and Data Analysis

The gametophytes of *A. flabellulatum* grew slowly. In order to obtain enough sequencing materials, RNA extraction was performed from gametophytes cultured for 60 d in 0 M and 10^−6^ M BL. Each treatment was set up with 3 biological replicates for a total of 6 samples. The CTAB method was used to extract the total RNA of each sample, then mRNAs were enriched by oligo (dT) magnetic beads. The enriched mRNAs were fragmented, and the cDNA libraries were constructed by reverse transcription. The BGISEQ-500 platform was used for sequencing. After removing raw reads with low quality, joint contamination, and unknown base N content greater than 5% by SOAPnuke (v1.4.0, Shenzhen, China), the clean reads were obtained [29]. The above steps were completed by the Beijing Genomics Institute (BGI, Beijing, China). Moreover, the clean reads of the 6 samples were uploaded to the SRA database of NCBI with the accession number PRJNA786072.

#### 2.2.3. Quality Evaluation of Transcriptome Sequencing

The clean reads were aligned to the full-length transcripts (PRJNA733457) of *A. flabellulatum* gametophytes by Bowtie2 (v2.2.5, Baltimore, MD, USA) [30,31], and then a sequencing saturation analysis was performed. The expression levels of genes (calculated in fragments per kilobase of transcript per million mapped reads, FPKM) in each sample were counted by RSEM software (v1.2.8, Madison, WI, USA) [32]. According to the methods of Cai et al. [33], Pearson’s correlation coefficient analysis, the principal component analysis (PCA), and a cluster analysis were performed on all samples.

#### 2.2.4. Screening of Differentially Expressed Genes, GO, and KEGG Analysis

After removing the isoforms with average FPKMs < 0.5 in both treatments, the differentially expressed genes (DEGs) were screened under the conditions of |log_2_ FoldChange| ≥ 1 and *Q*-Values < 0.001. According to the methods of Cai et al. [30], all DEGs were annotated and classified by GO (http://www.geneontology.org/ (accessed on 24 April 2022)) and KEGG (http://www.genome.jp/kegg/ (accessed on 24 April 2022)) analysis [30]. After that, enrichment of the analysis was performed by using the Phyper package in R software (Auckland, New Zealand). GO and KEGG annotations of the DEGs were completed by BGI.

#### 2.2.5. Quantitative Real-Time Polymerase Chain Reaction

According to the methods of Cai et al. [33], 18 genes were selected for qRT-PCR (Appendix A). Three parallel tests were performed for each sample. The isoform_38906 (*Actin*) was used as the reference gene and the 2^-ΔΔCt^ method was used for relative quantification [34].

## 3. Results

### 3.1. Quality Evaluation of Transcriptome Sequencing

Eight concentrations of BL were set up to observe their effects on the growth of *A. flabellulatum* gametophytes. The results showed that the perimeters and the projected areas of *A. flabellulatum* gametophytes were inhibited significantly when the concentrations of BL were more than 10^−7^ M (Appendix A), which were consistent with the results by Wang et al. [10]. However, when the concentrations of BL were between 0 and 10^−8^ M, the gametophytes changed irregularly (Appendix A). Therefore, transcriptome sequencing was performed on six samples from 0 M and 10^−6^ M BL treatments (represented by CK and BL, respectively). The results showed that each sample generated at least 6.32 Gb data, with an average of 6.34 Gb data. The clean reads were obtained after quality filtering was performed on the raw reads, the acquisition rates were greater than 96%, wherein Q20 > 97% and Q30 > 92% (Appendix A). A sequencing saturation analysis showed that the increase of gene identification ratios in each sample tended to be flat when the read numbers were greater than 50 × 100 K, which indicated the sequencing data reached saturation (Appendix A).

The cluster and correlation analyses were performed on the six samples. The results showed that three biological replicates in each treatment were clustered into one branch (Appendix A), and the Pearson correlation coefficients between the samples of each treatment were greater than or equal to 0.97 (Figure 1A). Moreover, the principal component analysis (PCA) showed that CK and BL treatments were separated on PC1 (Figure 1B). In conclusion, the six samples were of high quality with reliable sequencing data, which could be further analyzed.

### 3.2. Screening of DEGs

A total of 210,415 isoforms were detected in the six samples, and the medians of expression levels were greater than 0.99 (Appendix A, Figure 2A). After removing the isoforms with average FPKMs of less than 0.5 in two treatments, 8394 differentially expressed genes (DEGs) were screened under the conditions of |log_2_ FoldChange| ≥ 1 and *Q*-Values < 0.001. Among them, 5081 (60.53%) DEGs were upregulated and 3313 (39.47%) DEGs were downregulated in the BL treatment (Appendix A, Figure 2B,C). The DEGs above were used for further analysis.

### 3.3. GO Analysis of DEGs

The 8394 DEGs were aligned to the GO database, among which 5275 DEGs were annotated, accounting for 62.84% (Appendix A). GO terms were classified into three categories, including the biological process (BP), cellular component (CC), and molecular function (MF). Among them, MF annotated the most DEGs (4184). In BP, 2146 DEGs were annotated to the “cellular process”, accounting for the highest proportion, followed by the “metabolic process” (1751). In CC, the DEGs were mainly classified into “cellular anatomical entity” (3495). In MF, most DEGs were annotated to “binding” (2635), followed by “catalytic activity” (2570). Additionally, in most GO terms, the upregulated DEGs in the BL treatment were in the majority (Appendix A, Figure 3A).

According to the GO enrichment results of DEGs, we showed the top 20 terms with low *Q*-values. Among them, “DNA binding” had the largest gene number (439), and the lowest *Q*-value, while “cilium” had the highest rich ratio (0.20). In addition, we found gene numbers in “thylakoid”, “photosynthesis”, and “photosystem” were all large, which were 350, 275, and 242, respectively (Appendix A, Figure 3B).

### 3.4. KEGG Analysis of DEGs

The 8394 DEGs were aligned to the KEGG database, among which 3146 DEGs were annotated, accounting for 37.48% (Appendix A). KEGG pathways were classified into five categories—“metabolism”, “genetic information processing”, “environmental information processing”, “cellular processes”, and “organismal systems”. Among them, “metabolism” annotated the most DEGs (1941) followed by “genetic information processing” (796). In “metabolism”, 1629 DEGs were annotated to “global and overview maps”, which had the largest gene number, followed by “carbohydrate metabolism” (614). In “genetic information processing”, 301 DEGs were annotated to “translation” and 272 DEGs were annotated to “transcription”. In “environmental information processing”, the DEGs were mainly annotated to “signal transduction” (349). In “cellular processes”, 212 DEGs were annotated to “transport and catabolism”, and in “organismal systems”, 182 DEGs were annotated to “environmental adaptation” (Appendix A, Figure 4A).

According to the KEGG enrichment results of DEGs, we showed the top 20 pathways with low *Q*-values. Among them, the pathways related to photosynthesis were “photosynthesis-antenna proteins”, “photosynthesis”, and “carbon fixation in photosynthetic organisms”, and the DEG numbers were 113, 155, and 93, respectively. In addition, we found two pathways related to plant hormones, which were “plant hormone signal transduction” and “BR biosynthesis”, and the DEG numbers were 203 and 33, respectively (Appendix A, Figure 4B).

### 3.5. The Expressions of Photosynthesis-Related Genes Were Inhibited

The KEGG analysis showed that the pathways of “photosynthesis-antenna proteins”, “photosynthesis”, and “carbon fixation in photosynthetic organisms” were enriched significantly, and the DEG numbers were all large. Therefore, we analyzed the DEG expressions in the above pathways.

The “photosynthesis-antenna proteins” pathway includes the LHCI complex and LHCII complex. The LHCI complex was composed of five subunits, (Lhca1–Lhca5) and the LHCII complex was composed of seven subunits (Lhcb1–Lhcb7). The analysis showed that, except for Lhca5 and Lhcb7, DEGs were detected in the rest of the subunits, and the vast majority of them were downregulated in the BL treatment (Appendix A, Figure 5A,B). The “photosynthesis” pathway includes “PSII complex”, “PSI complex”, “cytochrome b6/f complex”, “photosynthetic electron transfer chain”, and “f-type ATPase”. The analysis showed that DEGs were detected in 26 subunits—8 subunits (PsbO, PsbP, PsbQ, PsbR, PsbS, PsbW, PsbY, Psb27) in the “PSII complex”, 11 subunits (PsaA, PsaB, PsaD, PsaE, PsaF, PsaG, PsaH, PsaK, PsaL, PsaN, PsaO) in the “PSI complex”, 1 subunit (PetA) in the “cytochrome b6/f complex”, 3 subunits (PetE, PetF, PetH) in the “photosynthetic electron transfer chain”, and 3 subunits (γ, delta, b) in “f-type ATPase”. Except for PsbY, all DEGs in the rest of the subunits were downregulated in the BL treatment (Appendix A, Figure 6A,B). There were 76 DEGs in the “Calvin cycle”, among them, *RBCS*, *GAPA*, *TKT*, *SBPASE*, *PRK,* and *RPE* were downregulated in the BL treatment, while *RPIA* was upregulated (Appendix A). In addition, almost all of the DEGs in “chlorophyll biosynthesis” as well as most of the DEGs in “carotenoid biosynthesis” were downregulated in the BL treatment (Appendix A).

### 3.6. The Expressions of BR Synthase Genes Were Negatively Regulated by BL

According to the results of the KEGG enrichment, the pathways of “plant hormone signal transduction” and “BR biosynthesis” were significantly enriched. Therefore, the expressions of encoding genes in the two pathways were analyzed. The results showed that in “BR signal transduction”, multiple members were detected, except *BSU1*. It is noteworthy that the encoding genes of co-receptor BAK1, transcription factor BZR1/2, and response factor TCH4 were upregulated by BL (Appendix A). Upregulated expression of *TCH4* indicated that BR signaling was activated continuously [35]. In addition, we detected many BR synthase genes *CYP90B1*, *CYP724B1*, *CPD*, *CYP90C1*, *CYP90D1*, *Det2*, *D2*, *BR6ox1,* and *CYP92A6* in *A. flabellulatum* gametophytes, except *BR6ox2.* Among them, *CYP90B1, CYP90C1, CYP90D1*, *CPD,* and *BR6ox1* were downregulated in the BL treatment (Appendix A, Figure 7A,B). So, the five kinds of BR synthase genes were negatively regulated by BL in *A. flabellulatum* gametophytes.

### 3.7. Quantitative Real-Time Polymerase Chain Reaction

In order to prove the reliability of transcriptome data, 18 DEGs were selected for qRT-PCR—3 DEGs (isoform_78015 (*Lhca1*), isoform_68594 (*Lhca2*), isoform_80079 (*Lhcb5*)) involved in the “photosynthesis-antenna proteins” pathway, 9 DEGs (isoform_66365 (*PsbO*), isoform_83395 (*PsbP*), isoform_84990 (*PsbQ*), isoform_58512 (*PsbS*), isoform_104649 (*PsaE*), isoform_90542 (*PsaL*), isoform_97688 (*PsaO*), isoform_37486 (*γ*), isoform_67541 (*b*)) in the “photosynthesis” pathway, 2 DEGs (isoform_45822 (*ALDO*), isoform_266684 (*SBPASE*)) in the “carbon fixation in photosynthetic organisms” pathway, 2 DEGs (isoform_59932 (*CYP90C1*), isoform_304741 (*CYP92A6*)) in the “BR biosynthesis” pathway, 1 DEG (isoform_299999 (*TCH4*)) in “BR signaling”, and 1 DEG (isoform_33804 (*ACSF*)) in “chlorophyll biosynthesis”. The results of qRT-PCR showed that the expression trends of the above 18 DEGs were the same as the results of the transcriptome (Appendix A, Figure 8), which indicated the reliability of transcriptome data.

## 4. Discussion

In angiosperms, sprinkling with 10^−8^ M BL can promote the net photosynthetic rate of *S. lycopersicum* leaves [36]. After pretreating seeds of *Pisum sativum* with 10^−7^ M BL, the chlorophyll content of its seedlings significantly increased [37]. However, for the *bes1-D* seedlings of *A. thaliana* (BR signaling was constitutively activated), the chlorophyll content significantly decreased [38]. In spore plants, the chlorophyll and carotenoid contents significantly increased by applying 10^−8^ M BL to *Chlorella vulgaris* and 10^−10^–10^−5^ M BL to the green alga *Acutodesmus obliquus* [39,40]. In the green alga *A. obliquus*, the promotional role of 10^−6^ M BL was the greatest [40]. In gene expression, the expressions of *RBCS* and *RBCL* in the “Calvin cycle” were upregulated when 10^−7^ M BL was sprayed on *Cucumis sativus* leaves [41], while the expressions of *GLK1* and *GLK2* (transcription factors responsible for chloroplast development and photosynthetic gene activation), as well as *Lhcb* (such as *Lhcb1.4* and *Lhcb2.2*) were downregulated by BL and/or in bes1-D seedlings of *A. thaliana* [38]. In this study, almost all of the DEGs in “photosynthesis-antenna proteins”, “photosynthesis”, and “chlorophyll biosynthesis”, as well as most of the DEGs in the “Calvin cycle”, and “carotenoid biosynthesis” were downregulated after 10^−6^ M BL was applied to *A. flabellulatum* gametophytes. In conclusion, BL is involved in regulating photosynthesis and its related gene expressions in angiosperms and spore plants (including fern gametophytes).

Whether BL exists in plants will vary according to the plant species. Nowadays, CS rather than BL is detected in liverwort (*Marchantia polymorpha*), moss (*Physcomitrella patens*), Lycopodiopsida (lycophytes) (*Selaginella moellendorffii* and *Selaginella uncinata*), and 12 species of Polypodiopsida (true ferns), including *P. aquilinum*, *O. japonica*, *Lygodium japonicum*, etc. [12]. In plants, the existence of BR6ox2 determines whether CS can be further converted into BL [15,16,17]. It is speculated that BR6ox2 is not present in ferns [12]. The results of our study support this conjecture because BR6ox2 was not detected in *A. flabellulatum* gametophytes. Although it is indisputable that CS is present in spore plants, the content is generally lower than that in angiosperms. For example, the CS content in spore plants (maximum is 38 pg g^−1^ fresh weight) is lower than 29% of that in *A. thaliana* (134 pg g^−1^ fresh weight), 26% in *O. sativa* (151 pg g^−1^ fresh weight), and 1.5% in *Z. mays* (2580 pg g^−1^ fresh weight) [12]. This may be related to the low expressions of BR synthase genes (Appendix A). Additionally, our studies showed that the expressions of five kinds of BR synthase genes (*CYP90B1*, *CYP90C1*, *CYP90D1*, *CPD*, and *BR6ox1*) were negatively regulated by BL, which indicated that ferns (including gametophytes) also needed the regulatory mechanism for maintaining BRs homeostasis.

## 5. Conclusions

In this study, the genes of fern gametophytes responsive to BRs were identified for the first time by transcriptome sequencing. We found that the expressions of photosynthetic genes were widely inhibited by high concentrations of BL in *A. flabellulatum* gametophytes. Moreover, many BR synthase genes were detected, except *BR6ox2*, which may be the reason why CS rather than BL is present in ferns. Finally, the expressions of BR synthase genes were negatively regulated by BL in *A. flabellulatum* gametophytes.

## Figures and Tables

**Figure 1 genes-13-01061-f001:**
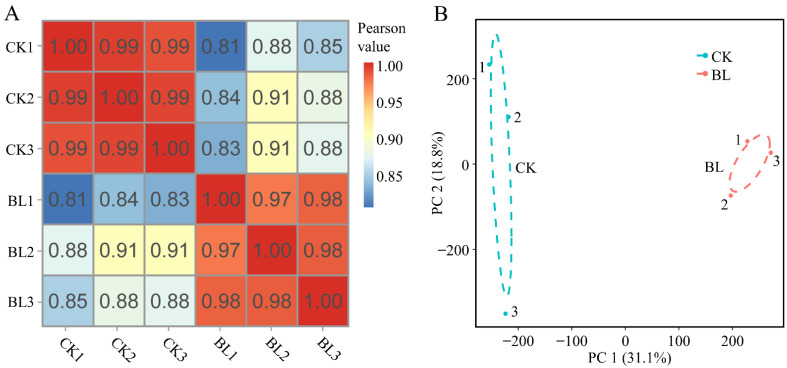
Pearson correlation coefficient analysis and principal component analysis. (**A**,**B**) The Pearson correlation coefficient analysis (**A**) and principal component analysis (PCA) (**B**) were performed among six samples of two treatments based on the gene expression data. The PCA diagram shows that CK and BL treatments were separated from each other on PC1.

**Figure 2 genes-13-01061-f002:**
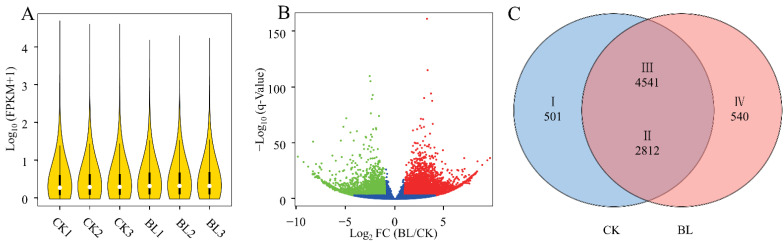
Screening of differentially expressed genes. (**A**) Distributions of gene expressions for the six samples. (**B**) DEGs were screened under the conditions of |log_2_ FoldChange| ≥ 1 and *Q*-Values < 0.001 (the red dots and the green dots indicate DEGs upregulated and downregulated in the BL treatment, respectively). (**C**) Venn diagram of the DEGs distributing in CK and BL treatments (I and II show the DEGs downregulated in the BL treatment; III and IV show the DEGs upregulated in the BL treatment. The DEGs belonging to I and IV were only expressed in CK and BL treatments, respectively; the same below).

**Figure 3 genes-13-01061-f003:**
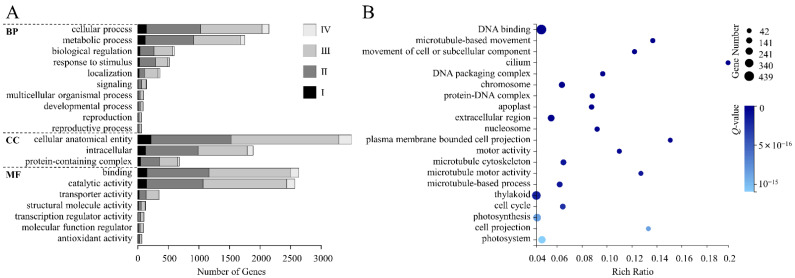
GO classification and enrichment. (**A**,**B**) GO functional classification (**A**) and enrichment (**B**) of DEGs. GO enrichment diagram shows the top 20 terms with low *Q*-values, the circle sizes indicate the DEG numbers, and the deeper colors indicate the lower *Q*-values.

**Figure 4 genes-13-01061-f004:**
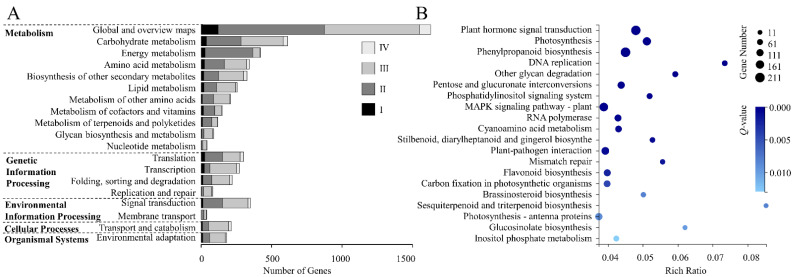
KEGG classification and enrichment. (**A**,**B**): KEGG pathway classification (**A**) and enrichment (**B**) of DEGs. The KEGG enrichment diagram shows the top 20 pathways with low *Q*-values; the circle sizes indicate the DEG numbers, the deeper colors indicate the lower *Q*-values.

**Figure 5 genes-13-01061-f005:**
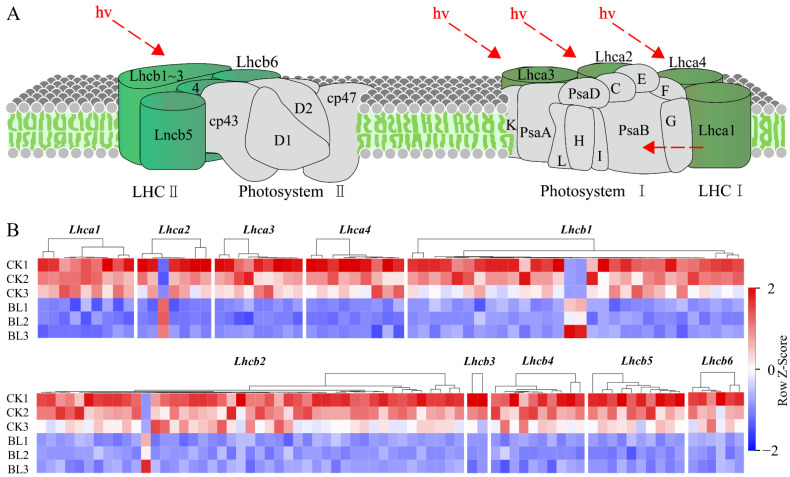
Almost all of the DEGs in the photosynthesis-antenna proteins pathway were downregulated in the BL treatment. (**A**) Photosynthesis-antenna proteins pathway (https://www.kegg.jp/pathway/map00196 (accessed on 24 April 2022)). (**B**) Expression heat maps of the DEGs in 10 subunits (Lhca1–Lhca4 and Lhcb1–Lhcb6 (accessed on 24 April 2022)) in the six samples, respectively.

**Figure 6 genes-13-01061-f006:**
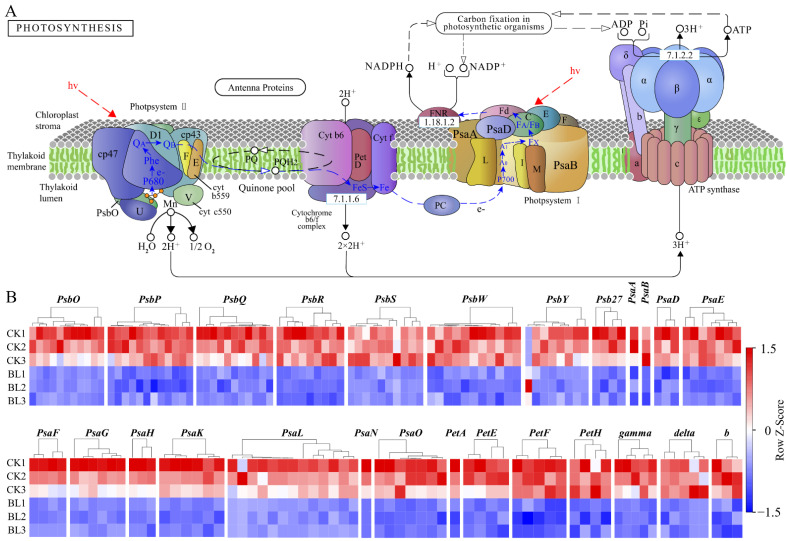
Almost all of the DEGs in the photosynthesis pathway were downregulated in the BL treatment. (**A**) Photosynthesis pathway (https://www.kegg.jp/entry/map00195 (accessed on 24 April 2022)). (**B**) Expression heat maps of the DEGs in 26 subunits—8 subunits (PsbO, PsbP, PsbQ, PsbR, PsbS, PsbW, PsbY, Psb27) in the “PSII complex”, 11 subunits (PsaA, PsaB, PsaD, PsaE, PsaF, PsaG, PsaH, PsaK, PsaL, PsaN, PsaO) in the “PSI complex”, 1 subunit (PetA) in the “cytochrome b6/f complex”, 3 subunits (PetE, PetF, PetH) in the “photosynthetic electron transfer chain”, and 3 subunits (γ, delta, b) in “f-type ATPase” in the six samples, respectively.

**Figure 7 genes-13-01061-f007:**
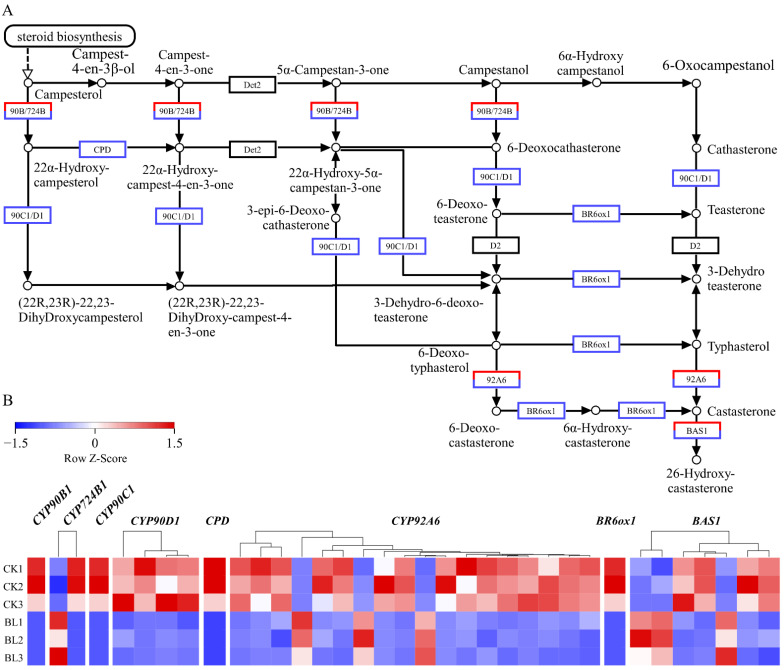
DEGs downregulated in the BR biosynthesis pathway were in the majority in the BL treatment. (**A**) The BR biosynthesis pathway (https://www.kegg.jp/entry/map00905 (accessed on 24 April 2022)). (**B**) Expression heat maps of the DEGs in *CYP90B1, CYP724B1, CYP90C1, CYP90D1*, *CPD*, *CYP92A6*, *BR6ox1,* and *BAS1* in the six samples, respectively.

**Figure 8 genes-13-01061-f008:**
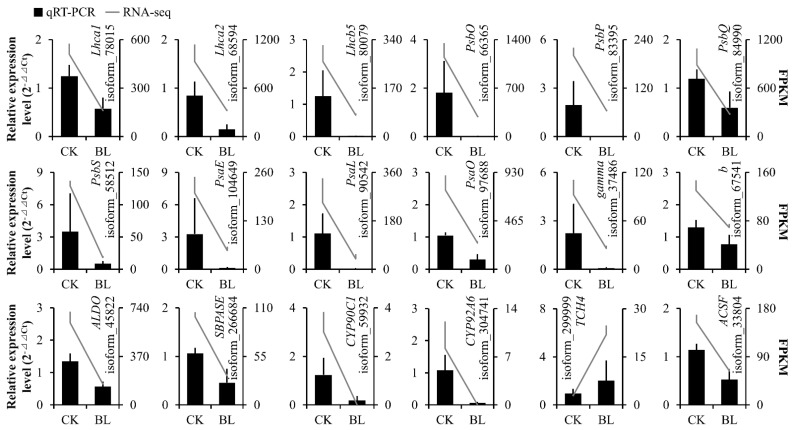
Quantitative real-time PCR verification. The expression trends of 18 DEGs were verified by qRT-PCR. The results of the qRT-PCR and transcriptome are shown by bar graphs and line charts, respectively.

## Data Availability

The following information was supplied regarding data availability: The clean reads are available at NCBI SRA (https://www.ncbi.nlm.nih.gov/sra/ (accessed on 24 April 2022)): PRJNA786072.

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
