# Peer review of "Excavation of Genes Responsive to Brassinosteroids by Transcriptome Sequencing in Adiantum flabellulatum Gametophytes"

_genes, 2022, doi:10.3390/genes13061061_

Round 1

Reviewer 1 Report

Dear Author,

I suggest to improve the introduction focusing in particular on the approach used for this study to identify BR and some exaample of study to identify BR by GO and KEGG.

Author Response

  1. I suggest to improve the introduction focusing in particular on the approach used for this study to identify BR and some example of study to identify BR by GO and KEGG.

Our response: Thanks for the comments. We have added two research examples in the introduction. The details are as follows:

Transcriptome sequencing has been widely used in screening of differentially expressed genes (DEGs) and pathway analysis. Huang et al. (2017), for example, treated Gerbera hybrida petals with BL, then used transcriptome sequencing, Gene Ontology (GO) and Kyoto Encyclopedia of Genes and Genomes (KEGG) analysis to identify the DEGs, which were annotated to multiple plant hormone signaling pathways. Wu et al. (2019) employed the methods above discovered that BL increased the contents of total glucosinolates and sulforaphane by upregulating the gene expressions related to glucosinolate core pathways in Brassica oleracea (Line 74-81).

Huang, G.; Han, M.; Yao, W.; Wang, Y. Transcriptome analysis reveals the regulation of brassinosteroids on petal growth in Gerbera hybrida. PeerJ 2017, 5, e3382, doi:10.7717/peerj.3382.

Wu, Q.; Wang, J.; Mao, S.; Xu, H.; Wu, Q.; Liang, M.; Yuan, Y.; Liu, M.; Huang, K. Comparative transcriptome analyses of genes involved in sulforaphane metabolism at different treatment in Chinese kale using full-length transcriptome sequencing. Bmc Genomics 2019, 20, 377, doi:10.1186/s12864-019-5758-2.

Reviewer 2 Report

This manuscript explores brassinosteroid-associated genes in Adiantum flabellulatum, using transcriptome sequencing with different bioinformatics annotation tools. This study reveals that a high concentration of brassinosteroids inhibits the photosynthetic genes and shows a large number of genes were associated with and regulated by brassinosteroids signaling. Overall, the work was rigorously performed, clearly presented, and should be of interest to a breadth of plant physiologists and pathologists interested in characterizing phytohormones signaling pathways. 

I have only minor suggestions for strengthening:

The title of the manuscript is very lengthy and hard to understand, and the authors should consider rewriting the title in a simple form.

The reference pattern is not uniform and not formatted according to author guidelines, e.g., 

According to guidelines, In the text, reference numbers should be placed in square brackets [ ] and placed before the punctuation; for example, [1], [1–3], or [1,3], not an author name and year. 

Reference numbers 17 and 24 were not formatted according to guidelines in the references list. Consider formatting those. 

Figure 8 is hard to understand because the font size is tiny. Could the authors represent this figure in an easy untestable way for the broad audience of the journal?

Author Response

Reviewer #2

  1. This manuscript explores brassinosteroid-associated genes in Adiantum flabellulatum, using transcriptome sequencing with different bioinformatics annotation tools. This study reveals that a high concentration of brassinosteroids inhibits the photosynthetic genes and shows a large number of genes were associated with and regulated by brassinosteroids signaling. Overall, the work was rigorously performed, clearly presented, and should be of interest to a breadth of plant physiologists and pathologists interested in characterizing phytohormones signaling pathways.

Our response: Thank you very much for your affirmation of the manuscript.

  1. The title of the manuscript is very lengthy and hard to understand, and the authors should consider rewriting the title in a simple form.

Our response: Thanks for the comments. We have revised the title of the manuscript as follows: Excavation of genes responsive to brassinosteroids by transcriptome sequencing in Adiantum flabellulatum gametophytes. (Line 1-2)

  1. The reference pattern is not uniform and not formatted according to author guidelines, e.g., According to guidelines, In the text, reference numbers should be placed in square brackets [ ] and placed before the punctuation; for example, [1], [1–3], or [1,3], not an author name and year.

Our response: Thanks. We have modified the reference format uniformly according to the reference format provided in the author guidelines. The specific amendments are shown in Line 397-536.

  1. Reference numbers 17 and 24 were not formatted according to guidelines in the references list. Consider formatting those.

Our response: Thanks. We have unified the formats of References 17 and 24. The amends are as follows:

Kwon, M.; Choe, S. Brassinosteroid biosynthesis and dwarf mutants. Journal of Plant Biology 2005, 48, 1, doi:10.1007/BF03030559. (Line 447-448)

Chen, Y.; Chen, Y.; Shi, C.; Huang, Z.; Zhang, Y.; Li, S.; Li, Y.; Ye, J.; Yu, C.; Li, Z.; Zhang, X.; Wang, J; Yang, H.; Fang, L.; Chen, Q. SOAPnuke: a MapReduce acceleration-supported software for integrated quality control and preprocessing of high-throughput sequencing data. Gigascience 2018, 7, 1-6, doi:10.1093/gigascience/gix120. (Line 490-494)

  1. Figure 8 is hard to understand because the font size is tiny. Could the authors represent this figure in an easy untestable way for the broad audience of the journal?

Our response: Thanks. We have adjusted the font size of Figure 8 to make it easier to identify. The modified Figure 8 is as follows:

(Line 327)

Reviewer 3 Report

Were any phenotypes observed at each BL concentration? 

Why was there a 60-day wait for RNA extraction? 

If BL is not present in ferns, why was BL used instead of CS to explore BR-regulated genes?

Author Response

Reviewer #3

  1. Were any phenotypes observed at each BL concentration?

Our response: Thanks. Supplementary Fig.1 showed the phenotypes of Adiantum flabellulatum gametophytes in eight different BL concentrations. The results showed that, the perimeters and the projected areas of A. flabellulatum gametophytes were inhibited significantly when the concentrations of BL were more than 10-7 M (Supplementary Fig. 1A-C). However, when the concentrations of BL were between 0 - 10-8 M, the gametophytes changed irregularly (Supplementary Fig. 1A).

We add this result to Line 156-158 in the revised manuscript.

Supplementary Fig. 1  A. flabellulatum gametophyte growth response to different BL concentrations

A-C: A. flabellulatum gametophytes cultured for 60 d in 0 M, 10-12 M, 10-11 M, 10-10 M, 10-9 M, 10-8 M, 10-7 M and 10-6 M BL respectively(A), as well as their perimeters (B) and projected areas (C). Different lowercase letters or capital letters in different groups indicated statistically significant differences (p < 0.05) or statistically extremely significant differences (p < 0.01).

  1. Why was there a 60-day wait for RNA extraction?

Our response: Thanks. The gametophytes of Adiantum flabellulatum grew slowly, and the gametophytes in high concentration BL treatment grew more slowly. In order to obtain enough sequencing materials, we selected the gametophytes cultured for 60 d for RNA extraction. (Line 113-114)

  1. If BL is not present in ferns, why was BL used instead of CS to explore BR-regulated genes?

Our response: Thanks. There are two reasons why we use BL to explore BR-regulated genes:

Brassinosteroids (BRs) are a class of polyhydroxysteroid plant hormones. Nowadays, more than 70 analogs have been found, among which brassinolide (BL) is the most active form (Line 48-50). The BR signal is transduced by receptor-like protein kinases BRI1/BAK1-mediated pathway, in which BRI1 and BAK1 are the receptor and co-receptor of BRs, respectively (Li et al. 2002; Nam and Li 2002) (Line 52-54).

BL is the most commonly used drug in the study of BR regulating plant development, and it is also commonly used in Oryza sativa, Zea mays, Pteridium aquilinum, and other plants that only synthesize CS (Suzuki et al. 1995; Kim et al. 2000; Hong et al. 2003; Kim et al. 2005; Gómez-Garay et al. 2018; Xiao et al. 2020; Tian et al. 2021) (Line 81-83).

Li, J.; Wen, J.; Lease, K.A.; Doke, J.T.; Tax, F.E.; Walker, J.C. BAK1, an Arabidopsis LRR receptor-like protein kinase, interacts with BRI1 and modulates brassinosteroid signaling. Cell 2002, 110, 213-222, doi:10.1016/s0092-8674(02)00812-7.

Nam, K.H.; Li, J. BRI1/BAK1, a receptor kinase pair mediating brassinosteroid signaling. Cell 2002, 110, 203-212, doi:10.1016/s0092-8674(02)00814-0.

Kim, Y.-S.; Kim, T.-W.; Kim, S.-K. Brassinosteroids are inherently biosynthesized in the primary roots of maize, Zea mays L. Phytochemistry 2005, 66, 1000-1006, doi:10.1016/j.phytochem.2005.03.007.

Suzuki, H.; Fujioka, S.; Takatsuto, S.; Yokota, T.; Murofushi, N.; Sakurai, A. Biosynthesis of Brassinosteroids in Seedlings of Catharanthus roseus, Nicotiana tabacum, and Oryza sativa. Bioscience, Biotechnology, and Biochemistry 1995, 59, 168-172, doi:10.1271/bbb.59.168.

Tian, X.; He, M.; Mei, E.; Zhang, B.; Tang, J.; Xu, M.; Liu, J.; Li, X.; Wang, Z.; Tang, W.; Guan, Q.; Bu, Q. WRKY53 integrates classic brassinosteroid signaling and the mitogen-activated protein kinase pathway to regulate rice architecture and seed size. The Plant cell 2021, 33, 2753-2775, doi:10.1093/plcell/koab137.

Xiao, Y.; Zhang, G.; Liu, D.; Niu, M.; Tong, H.; Chu, C. GSK2 stabilizes OFP3 to suppress brassinosteroid responses in rice. Plant J 2020, 102, 1187-1201, doi:10.1111/tpj.14692.

Hong, Z.; Ueguchi-Tanaka, M.; Umemura, K.; Uozu, S.; Fujioka, S.; Takatsuto, S.; Yoshida, S.; Ashikari, M.; Kitano, H.; Matsuoka, M. A rice brassinosteroid-deficient mutant, ebisu dwarf (d2), is caused by a loss of function of a new member of cytochrome P450. The Plant cell 2003, 15, 2900-2910, doi:10.1105/tpc.014712.

Kim, S.K.; Chang, S.C.; Lee, E.J.; Chung, W.S.; Kim, Y.S.; Hwang, S.; Lee, J.S. Involvement of brassinosteroids in the gravitropic response of primary root of maize. Plant physiology 2000, 123, 997-1004, doi:10.1104/pp.123.3.997.

Yokota, T.; Ohnishi, T.; Shibata, K.; Asahina, M.; Nomura, T.; Fujita, T.; Ishizaki, K.; Kohchi, T. Occurrence of brassinosteroids in non-flowering land plants, liverwort, moss, lycophyte and fern. Phytochemistry 2017, 136, 46-55, doi:10.1016/j.phytochem.2016.12.020.

Gómez-Garay, A.; Gabriel y Galán, J.M.; Cabezuelo, A.; Pintos, B.; Prada, C.; Martín, L. Ecological Significance of Brassinosteroids in Three Temperate Ferns. In Current Advances in Fern Research, Fernández, H., Ed.; Springer International Publishing: Cham, 2018; pp. 453-466, doi:10.1007/978-3-319-75103-0_21.
